



# The impact of stratospheric aerosol intervention on the North Atlantic and Quasi-Biennial Oscillations in the Geoengineering Model Intercomparison Project (GeoMIP) G6sulfur experiment

Andy Jones[1], Jim M. Haywood[1,2], Adam A. Scaife[1,2], Olivier Boucher[3], Matthew Henry[2], Ben Kravitz[4,5], Thibaut Lurton[3], Pierre Nabat[6], Ulrike Niemeier[7], Roland Séférian[6], Simone Tilmes[8], and Daniele Visioni[9]

[1]Met Office Hadley Centre, Exeter, UK
[2]College of Engineering, Mathematics and Physical Sciences, University of Exeter, Exeter, UK
[3]Institut Pierre-Simon Laplace, Sorbonne Université/CNRS, Paris, France
[4]Department of Earth and Atmospheric Science, Indiana University, Bloomington, IN, USA
[5]Atmospheric Sciences and Global Change Division, Pacific Northwest National Laboratory, Richland, WA, USA
[6]CNRM, Université de Toulouse, Météo-France, CNRS, Toulouse, France
[7]Max Planck Institute for Meteorology, Hamburg, Germany
[8]Atmospheric Chemistry, Observations and Modeling Laboratory, National Center for Atmospheric Research, Boulder, CO, USA
[9]Sibley School for Mechanical and Aerospace Engineering, Cornell University, Ithaca, NY, USA

**Correspondence:** Andy Jones (andy.jones@metoffice.gov.uk)

**Abstract.** As part of the Geoengineering Model Intercomparison Project a numerical experiment known as G6sulfur has been designed in which temperatures under a high-forcing future scenario (SSP5-8.5) are reduced to those under a medium-forcing scenario (SSP2-4.5) using the proposed geoengineering technique of stratospheric aerosol intervention (SAI). G6sulfur involves introducing sulphate aerosol into the tropical stratosphere where it reflects incoming sunlight back to space, thus cooling the planet. Here we compare the results from six Earth-system models which have performed the G6sulfur experiment and examine how SAI affects two important modes of natural variability, the northern wintertime North Atlantic Oscillation (NAO) and the Quasi-Biennial Oscillation (QBO). Although all models show that SAI is successful in reducing global-mean temperature as designed, they are also consistent in showing that it forces an increasingly positive phase of the NAO as the injection rate increases over the course of the 21[st] century, exacerbating precipitation reductions over parts of southern Europe compared with SSP5-8.5. In contrast to the robust result for the NAO there is less consistency for the impact on the QBO, but the results nevertheless indicate a risk that equatorial SAI could cause the QBO to stall and become locked in a phase with permanent westerly winds in the lower stratosphere.



# 1 Introduction

Global warming has accelerated swiftly over the last decade with the last seven years being warmer than any preceding years in the climatological record (e.g., https://climate.nasa.gov/vital-signs/global-temperature/). Climate model simulations suggest continued global warming throughout the next decades irrespective of emissions associated with scenarios of future economic growth (known as shared socio-economic pathways or SSPs; O'Neill *et al.*, 2016). As a consequence, there is growing recognition that the global-mean temperature targets of 1.5 °C and 2 °C above pre-industrial agreed at the Paris 21$^{st}$ Conference of Parties are going to be extremely difficult to achieve under conventional mitigation scenarios (e.g., Rogelj *et al.*, 2016; Millar *et al.*, 2017; IPCC, 2018; Tollefson, 2018). There is also a growing body of evidence that climate-induced damages frequently scale exponentially rather than linearly with temperature for metrics such as the frequency of extreme precipitation (Myhre *et al.*, 2019), heatwaves (Christidis *et al.*, 2015), droughts (Samaniego *et al.*, 2018) and possibly tropical cyclones (Knutson *et al.*, 2020). A further concern is that warming levels could be reached whereby key elements of the climate system such as the Amazon rainforest or the West Antarctic ice sheet could change dramatically in response to only a small additional warming (Lenton *et al.*, 2019; Wunderling *et al.*, 2021). These concerns have led to calls for research into less-conventional mitigation strategies (e.g., Royal Society, 2009; MacMartin *et al.*, 2018; NAS, 2021). These include proposals to remove greenhouse gases from the atmosphere (frequently called carbon dioxide removal) and proposals to either block sunlight from reaching the planet or to increase the albedo of the planet to reflect more sunlight out to space (frequently called solar radiation management, SRM).

Among the most prominent of the proposed SRM strategies in the scientific literature is stratospheric aerosol intervention (SAI) which proposes injecting aerosols or their precursors into the stratosphere where their atmospheric lifetime is considerably extended compared with that in the troposphere and where the aerosols can reflect sunlight back to space, thereby cooling the planet (Royal Society, 2009; Lawrence *et al.*, 2018; NAS, 2021). The injection material that has most frequently been studied is sulphur dioxide, in part because of work to understand and model the climatic impacts of large volcanic eruptions which periodically inject millions of tonnes of sulphur dioxide into the stratosphere. The resultant stratospheric sulphate aerosol from both large eruptions such as that of Mount Pinatubo in 1991 and the combined impacts of numerous smaller eruptions that took place over the period 2005-2012 have been shown to cool the climate (e.g., Soden *et al.*, 2002; Haywood *et al.*, 2014; Santer *et al.*, 2014; Schmidt *et al.*, 2018).

Two important natural modes of variability in the atmosphere are the North Atlantic Oscillation (NAO; e.g., Hurrell, 1995; Rodwell *et al.*, 1999) and the Quasi-Biennial Oscillation (QBO; e.g., Lindzen and Holton, 1968; Baldwin *et al.*, 2001). The NAO is defined by the anomaly in the mean sea level pressure (MSLP) between northern and sub-tropical regions of high variability in the Atlantic; locations in Iceland and the Azores are frequently used as these have the benefit of well-established long records of MSLP and are close to the centres of action of the dipolar NAO. The positive phase of the NAO is associated with an increase in the pressure gradient between the two regions which is associated with a strengthening of the jet-stream and a northward shift of the Atlantic storm track (e.g., Shindell *et al.*, 2004). Zanardo *et al.* (2019) performed an observational





analysis which showed that a positive phase of the NAO during Northern Hemisphere winter (defined throughout this study as December-February, DJF) is clearly associated with catastrophic flooding events in northern Europe. Similar positive precipitation anomalies were found in northern Europe during the positive phase of the NAO in DJF by López-Moreno and Vicente-Serrano (2008) and Casanueva *et al*. (2014) and higher levels of extreme precipitation were found by Scaife *et al*. (2008); the latter studies and Trigo *et al*. (2004) also report a concurrent reduction in precipitation in southern Europe. There

has been much debate as to whether aerosols from explosive volcanic eruptions which inject material into the stratosphere could cause warmer winters over Eurasia by affecting the NAO (e.g., Robock and Mao, 1992; Stenchikov *et al*., 2002, Fischer *et al.,* 2007, Marshall *et al*., 2009). As well as absorbing outgoing terrestrial radiation, stratospheric aerosols absorb sunlight in the near infra-red region of the solar spectrum and also increase the mean photon path and therefore absorption of solar radiation in ozone-absorbing bands. In northern wintertime, this absorption leads to heating in the sunlit parts of the

stratosphere at lower latitudes, thereby strengthening the temperature gradient and hence the polar vortex and inducing a positive phase of the NAO, increasing precipitation in northern Europe while decreasing it in southern Europe (Shindell *et al*., 2004; Scaife *et al*. 2008; Marshall *et al.,* 2009). However, any induced positive anomaly in the NAO subsequent to volcanic eruptions was found to be under-represented in Coupled Model Intercomparison Project phase 5 models (Driscoll *et al*., 2012) and there are arguments that inter-annual variability dominated over any induced response in a recent study of the 1991 Mount

Pinatubo eruption (Polvani *et al*., 2019). Although there are differences between volcanic eruptions and SAI, the most obvious of which is that explosive volcanic eruptions are sporadic while SAI is most frequently modelled using continuous emissions, there are obvious similarities between them and so two recent studies have examined the possible effects of SAI on the NAO. Jones *et al*. (2021) used ensembles of three simulations from two Earth-system models while Banerjee *et al*. (2021) used a 20-member ensemble from a single model; both concluded that SAI could induce a significant positive anomaly in the NAO.

The QBO is characterised by downward-propagating easterly and westerly wind regimes in the equatorial stratosphere with a period of around 28 months (Baldwin *et al*., 2001) and is caused by the interaction of a broad spectrum of vertically propagating gravity waves with the mean flow (Lindzen and Holton, 1968). The reversal of the equatorial flow in the stratosphere is associated with larger-scale changes in the dynamics of the stratosphere and hence the transport of chemical species out of the tropical stratosphere to higher latitudes. The QBO also influences weather at the surface through its influence

on the polar vortices, tropospheric jet-streams and storm-tracks (e.g., Holton, 1980; Kidston *et al*., 2015; Wang *et al*., 2018) and the phase of the QBO is known to influence the zonal and meridional transport of stratospheric volcanic aerosols from equatorial injections (e.g., Jones *et al*., 2016). Aquila *et al*. (2014) investigated the impact of equatorial stratospheric aerosol from SAI on the QBO and showed that progressively larger stratospheric sulphate aerosol concentrations increased the period of the QBO and could, if of sufficient magnitude, cause the QBO to stall, resulting in a permanent westerly phase. Jones *et al*.

(2016) examined the impacts of various aerosols as candidate SAI particles and found that those aerosols which absorbed more in the solar spectrum were the most effective in locking the QBO into a permanent westerly phase through their impacts on stratospheric temperatures.





The Geoengineering Model Intercomparison Project (GeoMIP) has been established for over a decade (Kravitz *et al*., 2011) and provides the most comprehensive multi-model assessment of the effects of SRM to date (e.g., Kravitz *et al*., 2013; Tilmes *et al*., 2013; Kravitz *et al*., 2021). In addition to several stand-alone model simulations (e.g., Tilmes *et al*. 2016; Jones *et al*., 2018), multi-model experiments with state-of-the-art climate models have progressed from relatively simple scenarios where the solar constant is reduced to offset an instantaneous quadrupling of carbon dioxide (e.g., Kravitz *et al*., 2013), to more policy-relevant experiments in the most recent phase of GeoMIP (Kravitz *et al*., 2015) which is aligned with the Coupled Model Intercomparison Project phase 6 (CMIP6; Eyring *et al*., 2016). In GeoMIP experiment G6sulfur, simulated global-mean temperature in a high-forcing scenario is reduced to the level of a medium-forcing scenario by the deployment of SAI geoengineering. The impacts on geographic temperature and precipitation distributions in these simulations have been shown to differ significantly from simulations which achieve the same global-mean temperature goal simply by reducing the solar constant (Jones *et al*., 2021; Visioni *et al*., 2021).

This study extends the work of Jones *et al*. (2021) on the possible impact of SAI geoengineering on the NAO by using a wider range of GeoMIP models. We also investigate the effect of SAI on the QBO in these models to try to obtain a more general view of impacts than that provided by previous single-model studies. Section 2 provides a description of the experimental design, Section 3 presents the results and Section 4 a discussion and conclusions.

## 2 Experiment Description

The GeoMIP G6sulfur experiment aims to alter simulations based on ScenarioMIP high-forcing scenario SSP5-8.5 (O'Neill *et al*., 2016; experiment ssp585) to follow the evolution of medium-forcing scenario SSP2-4.5 (experiment ssp245) over the period 2020-2100 by including gradually increasing amounts of SAI in the G6sulfur simulations. The criterion for comparing the G6sulfur and ssp245 simulations was initially defined by Kravitz *et al*. (2015) in terms of radiative forcing but this was subsequently altered to specify that the decadal global-mean near-surface air temperatures of the two simulations should be the same to within 0.2 °C.

We examine the impact of SAI in the six models which have performed the G6sulfur simulations to date (Table 1); more information can be found in Visioni *et al*. (2021) and references therein. Kravitz *et al*. (2015) were not prescriptive about how SAI should be implemented in the models as the details depend on each model's capabilities, resulting in different approaches that can be grouped into two basic categories. Three models injected $SO_2$ into the stratosphere and then interactively modelled the subsequent gas- and aerosol-phase processes: CESM2-WACCM injected $SO_2$ on the equator at the dateline at an altitude of 25 km, while IPSL-CM6A-LR and UKESM1-0-LL followed the suggestion of Kravitz *et al*. (2015) and injected along a line from 10° N to 10° S on the Greenwich meridian at 18-20 km. The other three models used prescribed aerosol optical depth (AOD) distributions: CNRM-ESM2-1 used a distribution provided by GeoMIP (from the G4SSA experiment; Tilmes *et al*., 2015) while MPI-ESM1-2-LR and MPI-ESM1-2-HR used distributions from simulations detailed in Niemeier and Schmidt (2017) and Niemeier *et al*. (2020).





Kravitz *et al*. (2015) specified that the G6sulfur experiments should consist of a three-member ensemble but not all models were able to provide three members; the ensemble size and realisation identifiers of the simulations available from each model are given in Table 1. In the analyses presented here we have used only those ssp245 and ssp585 ensemble members which correspond to each model's G6sulfur ensemble members, even if there are more ensemble members available from the ssp245 or ssp585 experiments. The only exception is MPI-ESM1-2-HR for which only two ensemble members are available

for ssp245 and ssp585 but three from G6sulfur.

## 3  Results

Where the results for each model are presented separately we show the model's ensemble mean. Where multi-model means are presented, each model's ensemble mean was used to construct the multi-model mean and the data were re-gridded to the resolution of the highest resolution model (MPI-ESM1-2-HR) before averaging.

### 3.1  SAI Cooling

All models were successful in reducing the global-mean temperatures in the G6sulfur simulations from the levels of ssp585 to within 0.2 °C of those in ssp245 (see Visioni *et al*., 2021, for details). Figure 1 shows each model's 2081-2100 ensemble mean difference in near-surface air temperature between G6sulfur and ssp585, indicating the amount of cooling required from SAI in the different models. There is considerable spread, with one group (CNRM-ESM2-1, MPI-ESM1-2-LR and MPI-ESM1-2-

HR) requiring a temperature reduction of  ~1.5 °C to cool ssp585 to ssp245 levels by the end of the century, while the other group (CESM2-WACCM, IPSL-CM6A-LR and UKESM1-0-LL) require a temperature reduction of ~2.5 °C by the end of the century. This demonstrates that as well as differences in the way the models simulate the impacts of SAI there are also considerable differences in the models' climate sensitivities and thus in the amount of warming they produce under a given scenario, often tied to their representation of clouds and how they respond in the future (Zelinka *et al*. 2020).

### 3.2  Impact on the NAO

Here we follow the approach of Stephenson *et al*. (2006) and Baker *et al*. (2018) in defining the boreal wintertime NAO as the mean DJF difference in area-mean sea-level pressure between two regions: one bounded by 90° W - 60° E, 20° N - 55° N, the other by 90° W - 60° E, 55° N - 90° N. The evolution of this pressure difference in the six models for ssp245 and G6sulfur is shown in Fig. 2; note that the y-axis range (10 hPa) is the same for all models even though the absolute values differ. Despite

considerable variability, all models agree in showing little systematic change in this measure of the NAO in ssp245 over the 2021-2100 period, the multi-model mean gradient of the straight-line fit being +0.02 hPa decade$^{-1}$ (range -0.03 to +0.16 hPa decade$^{-1}$). In contrast, all models in G6sulfur exhibit a positive trend in the pressure difference with a multi-model mean





gradient of +0.63 hPa decade$^{-1}$ (range +0.37 to +1.11 hPa decade$^{-1}$). The ranges of the slopes from the two sets of simulations do not overlap, clearly showing that G6sulfur, while maintaining global-mean temperature at the same level as ssp245, also

causes the wintertime NAO to become increasingly more positive through the century. However, the inclusion of SAI is obviously not the only difference between G6sulfur and ssp245. G6sulfur also includes the higher levels of greenhouse gases (GHG) and other changes of the ssp585 experiment whose warming effects the SAI is designed to offset. In order to assess whether the NAO changes seen in G6sulfur are due to SAI, data are required from simulations without SAI, which have the same GHG levels as G6sulfur, but without the confounding factor of a difference in temperature, as studies of scenarios with

warming levels similar to ssp585 have been found to affect the NAO (e.g., Tsanis and Tapoglou, 2019). These criteria are fulfilled by the GeoMIP experiment G6solar (Kravitz *et al*., 2015) which is parallel to G6sulfur but achieves the cooling from ssp585 to ssp245 levels by the highly idealised method of reducing the specified solar output. The multi-model mean gradient of the NAO timeseries in G6solar is -0.04 hPa decade$^{-1}$ (range -0.20 to +0.13 hPa decade$^{-1}$) which is similar to the gradient in ssp245, supporting the conjecture that SAI is responsible for the change in the NAO in G6sulfur.

The distributions of Northern Hemisphere DJF mean temperature differences between G6sulfur and ssp245 for 2081-2100 are shown in Fig. 3. While varying in degree, all models show a clear warming over northern Eurasia consistent with the positive NAO anomaly (Hurrell, 1995; Shindell *et al*., 2004), although the warming is still less than in ssp585. They also show cooling over the Labrador Sea and warming over the eastern USA, again as expected from a long term positive shift in the NAO (Scaife *et al.,* 2005), although the picture is less consistent for differences over North America (Banerjee *et al*., 2021;

Jones *et al*., 2021). The corresponding differences in Northern Hemisphere wintertime precipitation are shown in Fig. 4. The models are consistent in showing increases in precipitation in northern Europe and reductions in southern Europe as observed for a positive phase of the NAO (Trigo *et al*., 2004; Scaife *et al*., 2008; Casanueva *et al*., 2014) but the boundary between these two regimes varies according to model; as with temperature, there is again less consistency over North America. The degree of inter-model consistency is indicated in Fig. 5 which shows the multi-model mean difference in 2081-2100 mean DJF

temperature and land precipitation rate between G6sulfur and ssp245; points where two thirds of the models agree on the sign of the change are stippled. Multi-model means are frequently found to be a better representation of reality than a single model (Tebaldi and Knutti, 2007) but the inter-model consistency in showing a forced positive phase of the wintertime NAO indicates in G6sulfur suggests that this result is robust.

        The comparisons presented so far represent the differences between two possible worlds with the same global-mean

temperature (that of the end of the century under scenario SSP2-4.5) but in which one world uses SAI (G6sulfur) and the other does not (ssp245). We now examine precipitation over Europe to compare changes from present-day under scenario SSP5-8.5 plus SAI (i.e. G6sulfur) with the situations under scenarios SSP5-8.5 and SSP2-4.5. Figure 6(a) shows the multi-model-mean differences in DJF mean land precipitation rate between end-of-century in ssp245 (mean over 2081-2100) and present-day (mean over 2011-2030) which shows a general increase in precipitation over most of central and northern Europe and a slight

reduction over southern Europe; these changes are amplified in the high-emissions ssp585 experiment (Fig. 6b). The use of SAI in G6sulfur ameliorates the increase in precipitation over northern Europe that occurs in ssp585 while exacerbating the



reduction in precipitation over southern Europe (Fig. 6c). This reduction is especially marked over the Iberian Peninsula where the multi-model mean wintertime precipitation is reduced by 6.7 % in ssp585 but by 22.3 % in G6sulfur compared with present-day, with corresponding reductions of 11.1% and 29.1% in precipitation-minus-evaporation. This would be of great concern given the projected strong drying trend and water scarcity in this region (e.g. Molina *et al*., 2020; Perkins-Kirkpatrick *et al*., 2020). So although G6sulfur is successful in maintaining global-mean temperature at ssp245 levels despite ssp585 levels of GHGs, the changes in European precipitation from the present are clearly not maintained at ssp245 levels.

### 3.3 Impact on the QBO

The impact of SAI on the QBO in single or pairs of models has been studied for a number of years (e.g., Aquila *et al*., 2014; Jones *et al*., 2016; Richter *et al*. 2017; Tilmes *et al*. 2018a; Niemeier *et al*. 2020; Franke *et al*., 2021). These studies indicate that heating of the lower stratosphere by SAI aerosols is the main factor impacting the QBO, with sufficient SAI affecting the thermal wind balance causing the QBO to stall and locking it into a permanent westerly phase (Aquila *et al*., 2014; Jones *et al*. 2016; Franke *et al*., 2021). Note that throughout this study we define the phase of the QBO as the direction of the zonal-mean equatorial winds at 30 hPa. The amount of SAI required to cause the QBO to shut down has been shown to be model dependent: it is affected by a model's handling of aerosol and aerosol-radiation interactions and also by the treatment of stratospheric dynamics which affects the model's vertical advection (Niemeier *et al*., 2020). Both Richter *et al*. (2017) and Franke *et al*. (2021) have shown that the geographic location of the SAI is of crucial importance in the response of the QBO. Richter *et al*. (2017) found that moving the SAI location away from the equator made the QBO period decrease in their model, while Franke *et al*. (2021) found that an amount of SAI which caused the simulated QBO to shut down when applied at the equator had much less of an effect when applied instead at 30° N and 30° S.

When examining the impact of SAI on the QBO in G6sulfur it is instructive to know how well the different models simulate the QBO in the absence of SAI in ssp245. Figure 7 (a – f) shows the stratospheric zonal wind averaged between 5° S and 5° N from the first ensemble member (simulation identifier beginning 'r1' in Table 1) of each model's ssp245 ensemble over the period 2020-2099 (both MPI-ESM1-2-LR and MPI-ESM1-2-HR's r1 simulations lack data for 2100 in G6sulfur so this year is also omitted here). All but one of the models are able to simulate a QBO but there are various deficiencies in the simulations, especially in relation to the amplitude of the QBO between ~20 and ~100 hPa, issues which have been present since the first general circulation model simulations of the QBO (Scaife *et al*., 2000; Giorgetta *et al*., 2002) and continue in CMIP6 models (Richter *et al*., 2020). The evolution of stratospheric winds in the models may be compared with those from the ERA5 reanalyses shown in Fig. 7(g) (Hersbach *et al*., 2019); Table 2 summarizes the performance of each model in simulating the QBO when compared with ERA5. Figure 7 also shows that there are no obvious changes in the QBO over the course of the century in ssp245 and certainly no sign of the QBO shutting down in the absence of SAI despite rare disruptions to its regular cycling (e.g., Osprey *et al*., 2016).



Figure 8 shows the stratospheric winds from the first ensemble member (r1) of each model's G6sulfur simulation. A
single ensemble member is plotted rather than an ensemble mean as any QBO shutdown is likely to be a discrete event at a
given point in time and it is not meaningful to average across an ensemble whose members may simulate this process at
different times. Of the five models which simulate QBO-like behaviour in their ssp245 simulations, three (IPSL-CM6A-LR,
MPI-ESM1-2-HR and UKESM1-0-LL) clearly show a shutting down of the QBO when SAI is applied resulting in persistent
westerlies in the lower stratosphere by the end of the century. The tendency towards permanent westerlies is also evident in
MPI-ESM1-2-LR even though this model shows no QBO-like periodicity in either ssp245 or G6sulfur. In contrast, two models
(CESM2-WACCM and CNRM-ESM2-1) do not show such behaviour, with alternating easterly and westerly winds continuing
to the end of the century in these models. There is no correlation between simulating a QBO shutdown and the method of
implementing SAI: two models which use $SO_2$ injection simulate a shutdown while one does not and similarly with those
using prescribed AOD distributions. There is no correlation between the amount of stratospheric warming induced by SAI and
the shutting down of the QBO: for example, the QBO in MPI-ESM1-2-HR r1 shuts down at the start of the 2030s with a
maximum zonal-mean stratospheric warming of 3.1 °C (mean over 2031-2040) compared with ssp245, whereas the QBO in
CESM2-WACCM r1 is still present at the end of the century with a warming of 8.8 °C (mean over 2091-2100). The onset of
QBO shutdown in those models in which it occurs is also very variable, with dates ranging from ~2030 to 2060-2070. Similar
behaviour is seen in the other ensemble members (not shown) with only slight variation in the timing of any QBO shutdown.
It is not the purpose of this study to enter into an examination of the ability or otherwise of these models to accurately simulate
the QBO or an analysis of why it shuts down in some models under SAI. Rather it is to demonstrate that there is no consistent
guidance from these state-of-the-art climate models, even in a well-defined experiment such as G6sulfur, as to the impact of
SAI on the QBO. Nevertheless, despite the lack of consistency, these results suggest that there is significant risk that the form
of SAI examined here could terminate the QBO.

One of the possible effects of a shutdown of the QBO and the resulting persistent westerly phase is related to the fact
that transport of material out of the tropical lower stratosphere is modulated by the phase of the QBO and is reduced during
the westerly phase (Punge *et al*., 2009; Visioni *et al*., 2018). Consequently, it is possible that a persistent westerly phase would
lead to reduced transport of SAI aerosol out of the equatorial region. There may also be a slight counteracting effect whereby
increased confinement of the $SO_2$ and aerosol in the equatorial region during the westerly phase leads to increased aerosol
growth and faster gravitational settling (Visioni *et al*., 2018). Obviously the effect of a QBO shutdown in our simulations can
only be examined in those models which both simulate SAI interactively by $SO_2$ injection and have a QBO shutdown, namely
IPSL-CM6A-LR and UKESM1-0-LL. Figure 9 shows the evolution of the ratio of SAI AOD (inferred as the difference in
stratospheric AOD between G6sulfur and ssp245) in the subtropics (30° N - 30° S excluding 10° N - 10° S) to that in the SAI
injection zone (10° N - 10° S). The value of this ratio clearly differs between the two models but it shows no significant change
following QBO shutdown in either model, suggesting little impact on transport from the injection region.



## 4 Discussion and Conclusions

The results presented above support the conclusions of Jones *et al.* (2021) regarding the impact on the NAO in G6sulfur using a larger number of models. A robust agreement is found across the models that SAI in G6sulfur tends to induce a positive phase of the wintertime NAO leading to warmer and wetter winters over northern Eurasia with cooler and drier winters over
southern Europe compared with ssp245.

The spatial pattern of the multi-model mean precipitation changes over Europe in G6sulfur are very similar to observations of the impact of the positive phase of the NAO from the Global Precipitation Climatology Project (GPCP; Adler *et al.*, 2003) as shown in Fig. 10, although obviously this is not an exact comparison as the NAO response in G6sulfur is diagnosed at the end of the 21st century. Observations show that during the positive phase of the NAO areas of Iberia and parts
of southern Europe such as the Balkan peninsula and Anatolia show the most significant precipitation reductions while western areas of Norway and the north west of Scotland show the most significant precipitation increases (Fig. 10b), a pattern of changes which is largely reproduced in G6sulfur (Fig. 10c). This similarity between observations of the precipitation response to the positive phase of the NAO and the impact in G6sulfur gives additional confidence that the models are performing well. The issue of wintertime precipitation changes driven by changes in the phase of the NAO and the associated changes in the
strength and position of the North Atlantic storm-track are unlikely to be solved by injecting $SO_2$ using more sophisticated geographic injection strategies owing to the relatively long lifetime of the sulphate aerosols in the stratosphere during which they may be transported (and induce stratospheric heating) far from the injection site. Simpson *et al.* (2018) and Banerjee *et al.* (2021) both performed simulations using the Geoengineering Large Ensemble (Tilmes *et al.*, 2018b) where the injections were performed at latitudes of 30° N, 15° N, 15° S and 30° S and both studies found a very similar forced positive phase of the
DJF NAO to that found here in G6sulfur using equatorial injection. Simpson *et al.* (2018) performed further experiments where the stratospheric heating was enhanced and found a stronger impact on the positive phase of the DJF NAO and the associated precipitation patterns, suggesting that the absorption of solar radiation at wavelengths greater than ~1.3 µm by stratospheric sulphate (Dykema *et al.*, 2016) is the root cause of this response. Consequently, any form of large-scale deployment of SAI using sulphate aerosol is likely to affect the NAO in a manner similar to that described here. That only a small amount of
aerosol absorption in the tail-end of the solar spectrum could have such impacts suggests that any proposals utilising even small amounts of highly absorbing aerosol such as black carbon (e.g., Gao *et al.*, 2021) need to be treated with caution. As noted by Dykema *et al.* (2016), there are other candidate particles that absorb less solar radiation than sulphate that might be considered more suitable, but climate modelling research into the impacts of such particles is still in its infancy. Additionally, the use of these alternative particles may be compromised by coatings of sulphate from natural sources (McGrory *et al.*, 2021).
In contrast to the results regarding the NAO, there is no consensus as to impacts on the QBO in G6sulfur. One model has the QBO locking into a persistent westerly phase within a few years of the commencement of SAI, in three models this effect does not occur until mid-century, and two models do not show this behaviour at all. The reasons for this are the subject for future study, but the ability of models to accurately simulate the unperturbed QBO is an obvious place to start. Nevertheless,





despite the lack of consistency, the results indicate a clear risk that SAI by the equatorial injection of $SO_2$ could cause a
permanent shutdown of the QBO. We note that any impacts on the QBO may be reduced by injection strategies that do not
inject $SO_2$ directly at the equator but are optimized to reduce residual climate impacts by injecting at 30° N and 30° S (Franke
*et al.*, 2021). Results from two of the models suggest that, even if the QBO does shut down, the impact on the distribution of
stratospheric aerosols is likely to be small. However, a fixed westerly phase of the QBO could partly undermine the purpose
of SAI by further exacerbating European temperature and rainfall changes via teleconnections with the NAO (Andrews *et al.*,
2019) and it is recognised that further work in this area is needed.

Our strongest conclusion is that a forced positive phase of the DJF NAO and the associated shifts in precipitation
over Europe are likely to remain systemic problems for any SAI strategy, particularly for Iberia where the reduction in
wintertime precipitation in G6sulfur is more significant than in the high-end climate change (ssp585) scenario that climate
engineering was designed to mitigate. How such issues could be dealt with in terms of societal and economic remediation is
beyond the scope of this work.

*Code and data availability*.  All model data used in this work are available from the Earth System Grid Federation (WCRP,
2021).

*Author contributions*.  AJ and JMH led the analysis and wrote the manuscript with contributions from all co-authors.

*Competing interests*.  The authors declare that they have no conflict of interest.

*Acknowledgements*.  AJ would like to thank the Met Office team responsible for the *managecmip* software which greatly
simplified data discovery and handling, and Neal Butchart for useful comments. AJ, JMH and AAS were supported by the
Met Office Hadley Centre Climate Programme funded by BEIS and Defra. AJ and JMH would like to acknowledge funding
provided by SilverLining through its Safe Climate Research Initiative. JMH and MH were part-funded by the National
Environment Research Council Exeter-NCAR (EXTEND) collaborative development grant (NE/W003880/1). The IPSL-
CM6A-LR experiments were performed using the high-performance computing resources of TGCC under the allocations
2019-A0060107732 and 2020-A0080107732 (project gencmip6) provided by GENCI (Grand Equipement National de Calcul
Intensif). IPSL benefited from the French state aid managed by the ANR under the "Investissements d'avenir" programme with
the reference ANR-11-IDEX-0004-17-EURE-0006.  Support for BK was provided in part by the National Science Foundation
through agreement CBET-1931641, the Indiana University Environmental Resilience Institute, and the *Prepared for
Environmental Change* Grand Challenge initiative. The Pacific Northwest National Laboratory is operated for the US



Department of Energy by Battelle Memorial Institute under contract DE-AC05-76RL01830. RS and PN particularly acknowledge the support of the team in charge of the CNRM-CM climate model; supercomputing time was provided by the

Météo-France/DSI supercomputing center. RS thanks Christophe Cassou for his fruitful discussion on the paper. CNRM-ESM2-1 simulations were supported by the European Union's Horizon 2020 research and innovation program via the H2020 projects CRESCENDO (grant agreement Nᵒ· 641816) and CONSTRAIN (grant agreement Nᵒ· 820829). UN was supported by the Deutsche Forschungsgemeinschaft Research Unit VollImpact (FOR2820, grant Nᵒ· 398006378) and used resources of the Deutsches Klimarechenzentrum (DKRZ) granted by its Scientific Steering Committee (WLA) under project ID bm0550. We

acknowledge the Copernicus Climate Change Service (C3S) Climate Data Store (CDS) for the ERA5 data.

*Financial support.* Financial support for the individual authors is described in the Acknowledgements.

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



## Tables

**Table 1.** The models used in this study with details of the experiment ensembles: the ensemble size, the identifiers of the individual simulations and the experiment data reference.

| Model | G6sulfur ensemble | ssp245 ensemble | ssp585 ensemble | G6solar ensemble |
|---|---|---|---|---|
| CESM2-WACCM (Danabasoglu *et al.*, 2020; Gettelman *et al.*, 2019) | 2: r1i1p1f2, r1i1p1f2 (Danabasoglu, 2019a) | 2: r1i1p1f1, r2i1p1f1 (Danabasoglu, 2019b) | 2: r1i1p1f1, r2i1p1f1 (Danabasoglu, 2019c) | 2: r1i1p1f1, r1i2p1f1 (Danabasoglu, 2019d) |
| CNRM-ESM2-1 (Séférian *et al.*, 2019a) | 3: r1i1p1f2, r2i1p1f2, r3i1p1f2 (Séférian, 2019b) | 3: r1i1p1f2, r2i1p1f2, r3i1p1f2 (Voldoire, 2019a) | 3: r1i1p1f2, r2i1p1f2, r3i1p1f2 (Voldoire, 2019b) | 1: r1i1p1f2 (Séférian, 2020) |
| IPSL-CM6A-LR (Boucher *et al.*, 2020a; Lurton *et al.*, 2020) | 1: r1i1p1f1 (Boucher *et al.*, 2020b) | 1: r1i1p1f1 (Boucher *et al.*, 2019a) | 1: r1i1p1f1 (Boucher *et al.*, 2019b) | 1: r1i1p1f1 (Boucher *et al.*, 2019c) |
| MPI-ESM1-2-LR (Müller *et al.*, 2018) | 3: r1i1p1f1, r2i1p1f1, r3i1p1f1 (Niemeier *et al.*, 2019a) | 3: r1i1p1f1, r2i1p1f1, r3i1p1f1 (Wieners *et al.*, 2019a) | 3: r1i1p1f1, r2i1p1f1, r3i1p1f1(Wieners *et al.*, 2019b) | 3: r1i1p1f1, r2i1p1f1, r3i1p1f (Niemeier *et al.*, 2019b) |
| MPI-ESM1-2-HR (Müller *et al.*, 2018) | 3: r1i1p1f1, r2i1p1f1, r3i1p1f1 (Niemeier *et al.*, 2019c) | 2: r1i1p1f1, r2i1p1f1 (Schupfner *et al.*, 2019a) | 2: r1i1p1f1, r2i1p1f1 (Schupfner *et al.*, 2019b) | 3: r1i1p1f1, r2i1p1f1, r3i1p1f1 (Niemeier *et al.*, 2019d) |
| UKESM1-0-LL (Sellar *et al.*, 2019) | 3: r1i1p1f2, r4i1p1f2, r8i1p1f2 (Jones, 2019a) | 3: r1i1p1f2, r4i1p1f2, r8i1p1f2 (Good *et al.*, 2019a) | 3: r1i1p1f2, r4i1p1f2, r8i1p1f2 (Good *et al.*, 2019b) | 3: r1i1p1f2, r4i1p1f2, r8i1p1f2 (Jones, 2019b) |


**Table 2.** The QBO period and a qualitative summary of the QBO simulation in ssp245 by each model's r1 ensemble member (Fig. 7a-f) based on a comparison
with ERA5 (Fig. 7g). The QBO period is estimated from the number of easterly-to-westerly zonal wind transitions at 30 hPa during 2020-2099.

| Model | QBO period (months) | Comment |
|---|---|---|
| CESM2-WACCM | 16 | Period too short. Westerlies do not penetrate far enough downwards, ending at ~40 hPa when the ERA5 data show them reaching down to 100 hPa. |
| CNRM-ESM2-1 | 17 | Period too short. Westerlies terminate at ~60 hPa and the descent is too uniform (no stalling of the transition from westerlies to easterlies at ~40 hPa). |
| IPSL-CM6A-LR | 25 | Westerlies do not reach 100 hPa. There is also a lot of westerly activity at ~10 hPa which is not seen in ERA5. |
| MPI-ESM1-2-LR | N/A | No QBO-like periodicity present. |
| MPI-ESM1-2-HR | 29 | Westerlies do not penetrate to 100 hPa. There are also long periods of westerly winds at ~10 hPa which are not evident in ERA5. |
| UKESM1-0-LL | 29 | Duration of westerlies below ~40 hPa is rather too long. |



**Figures**

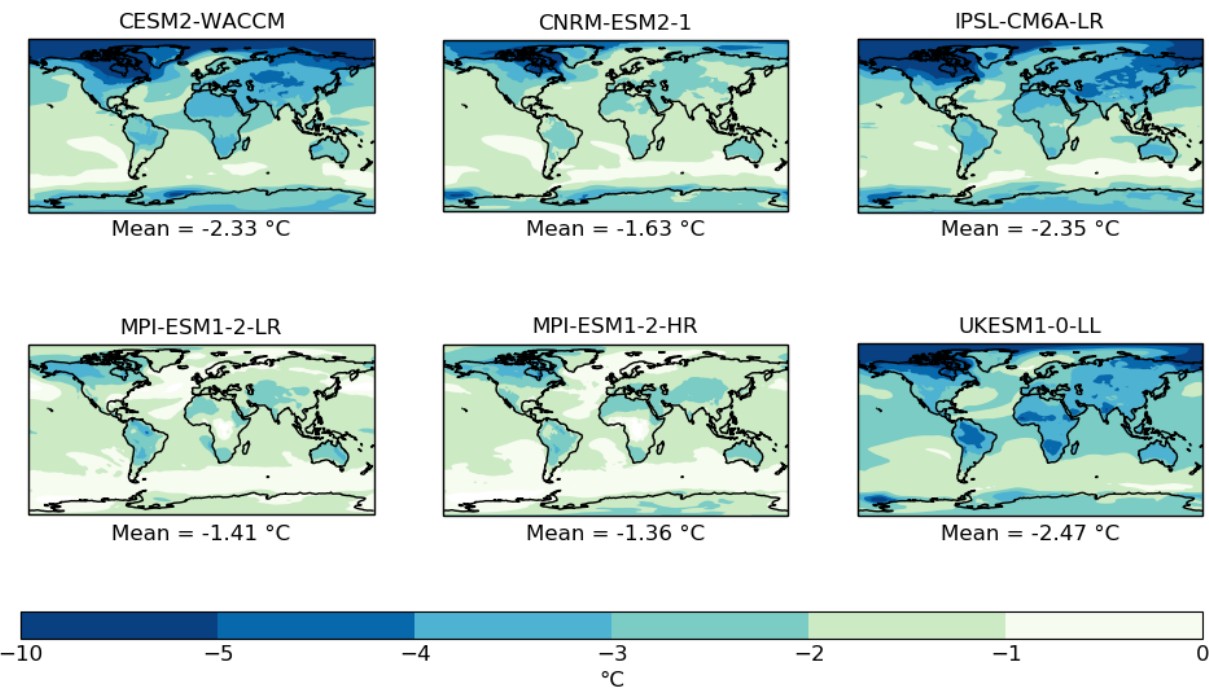

**Figure 1:** G6sulfur minus ssp585 difference in 2081-2100 annual mean near-surface air temperature (°C) for each model; all results are ensemble means.







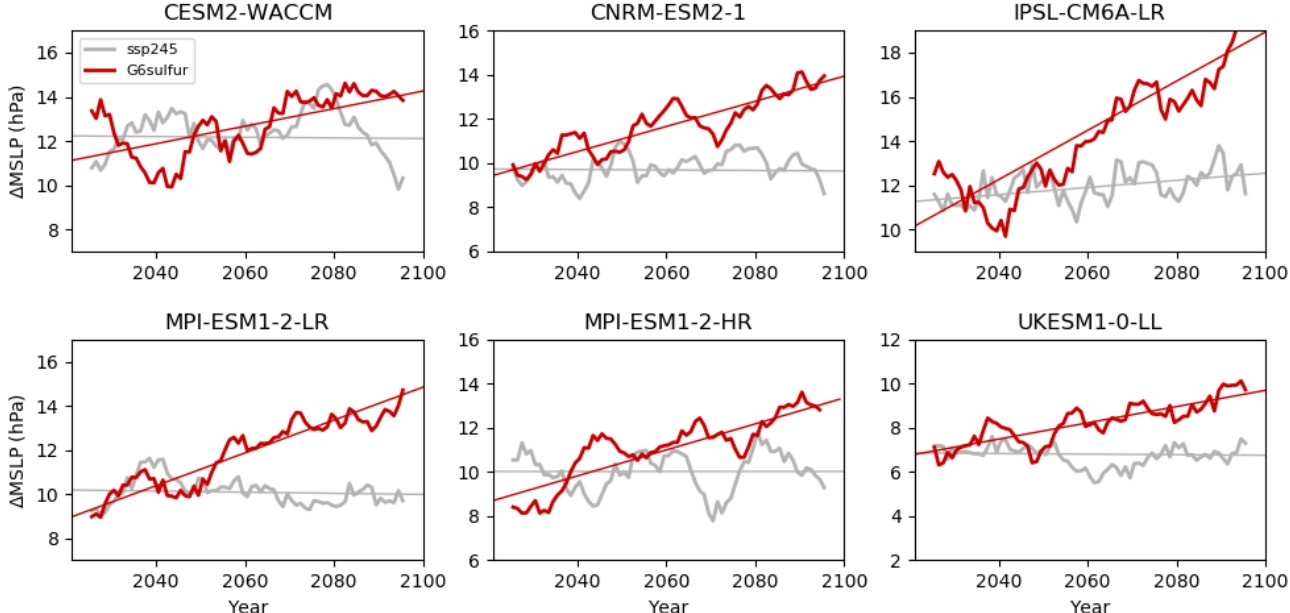

**Figure 2:** Evolution of the NAO, defined as the DJF mean sea-level pressure difference between regions bounded by 90° W - 60° E, 20° N - 55° N and 90° W - 60° E, 55° N - 90° N (hPa), for each model in experiments ssp245 (grey) and G6sulfur (red). All results are ensemble means and have been smoothed using a 10-year running mean; a least-squares straight line fit to each is also plotted.




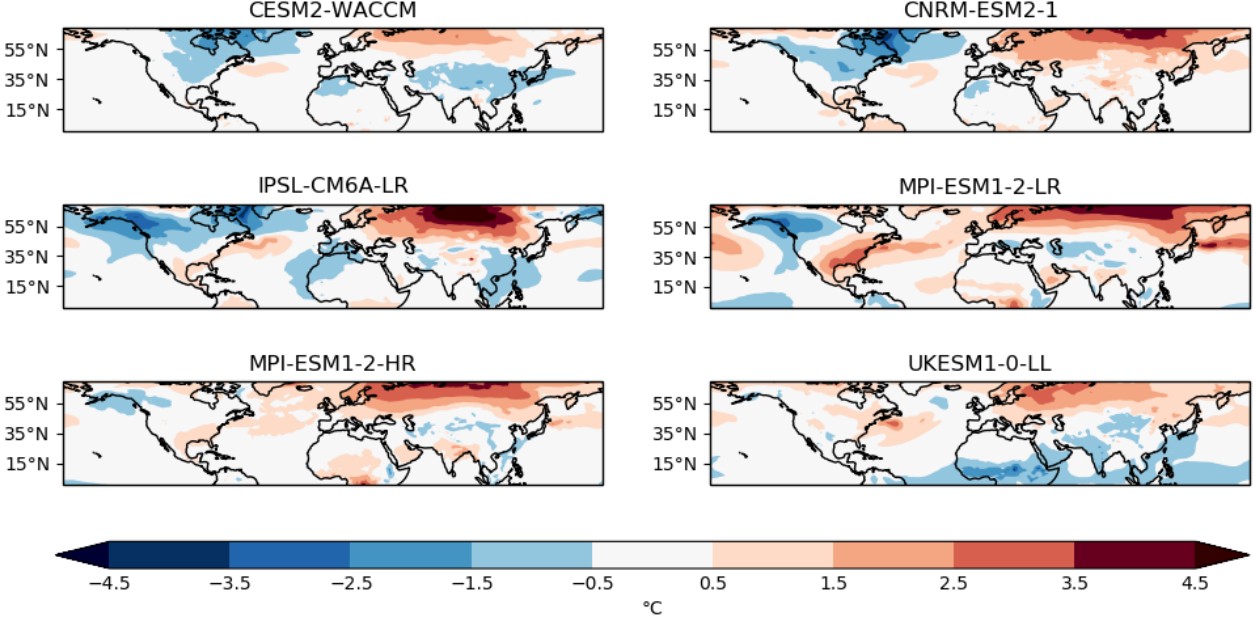

**Figure 3:** G6sulfur minus ssp245 difference in 2081-2100 mean DJF near-surface air temperature (°C) for each model. The area plotted replicates that presented by Jones *et al.* (2021; their Fig. 8) to concentrate on the area affected by the NAO.



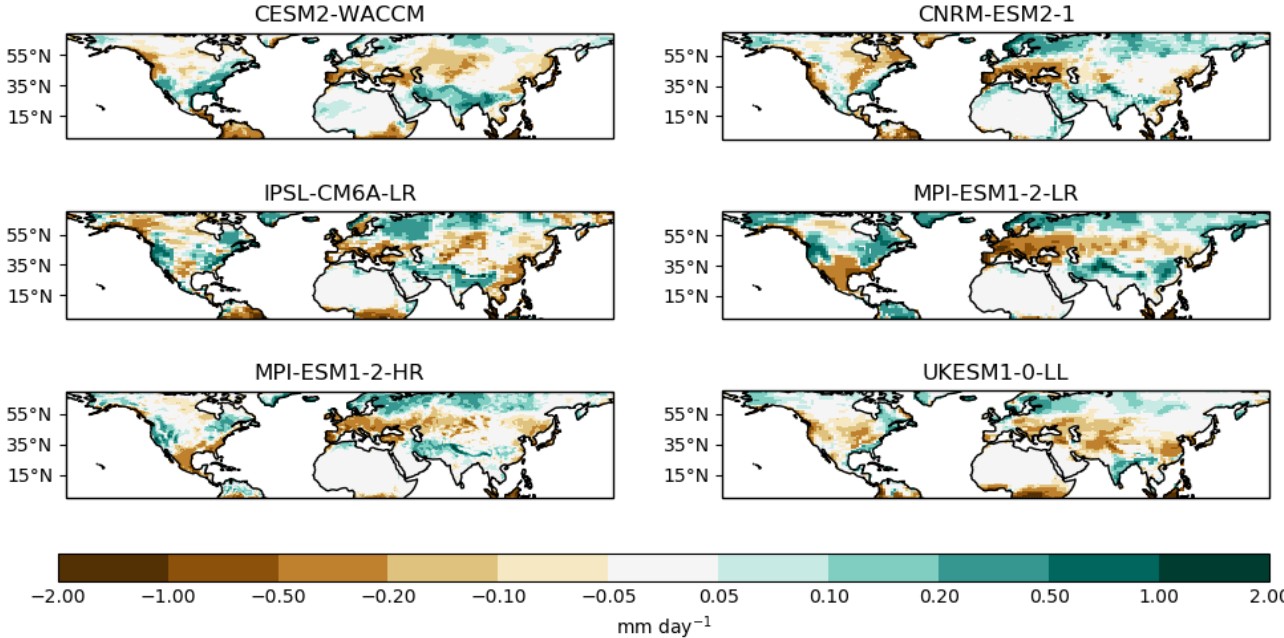

**Figure 4:** As Fig. 3 but for land precipitation rate (mm day[-1]).





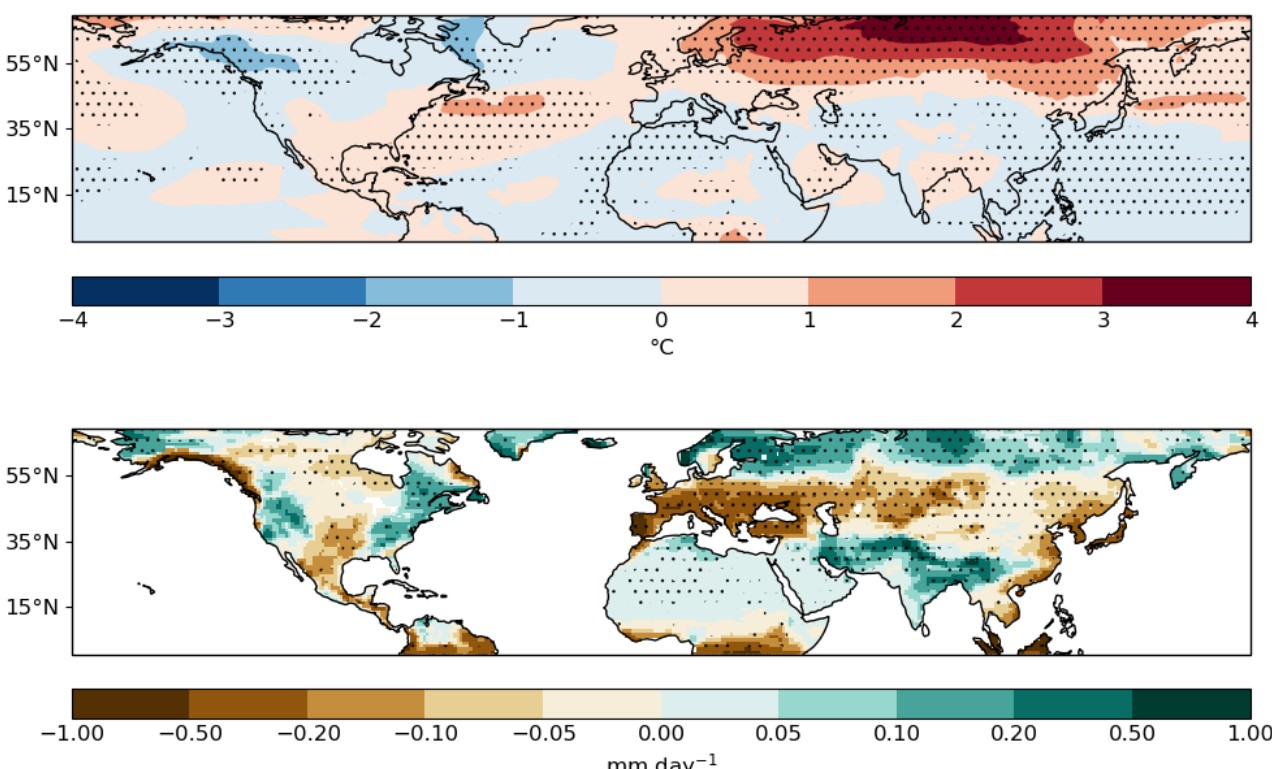

**Figure 5:** Multi-model mean difference in 2081-2100 DJF near-surface air temperature (°C, top) and land precipitation rate (mm day$^{-1}$, bottom) between G6sulfur and ssp245. Points where at least two-thirds of models (4 out of 6) agree on the sign of the difference are stippled.





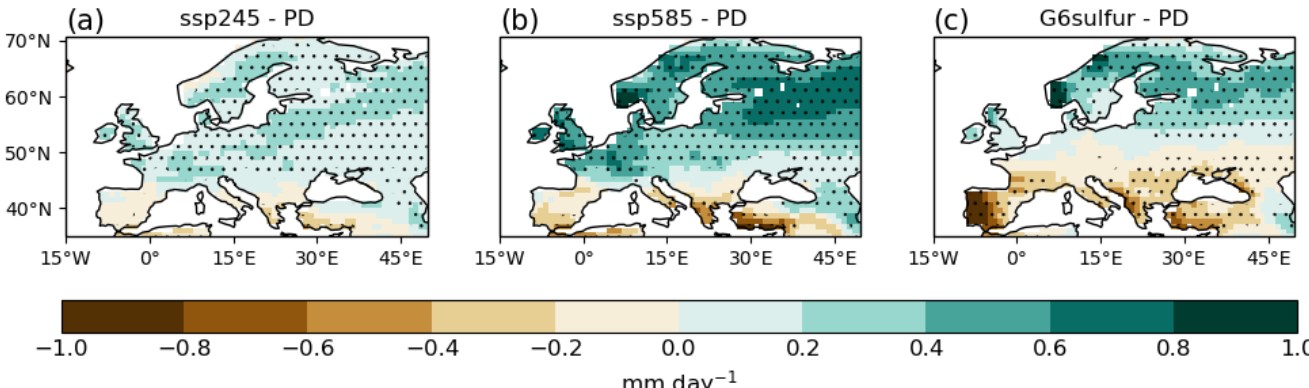

**Figure 6:** Multi-model mean difference in DJF mean land precipitation rate (mm day⁻¹) between ssp245 (2081-2100) and present day (PD; 2011-2030) **(a)**. Same as (a) but for the difference between ssp585 (2081-2100) and PD **(b)**. Same as (a) but for the difference between G6sulfur (2081-2100) and PD **(c)**. The PD data comprise years 2011-2014 from the CMIP6 historical simulations and years 2015-2030 from the corresponding ssp245 simulations. Points where at least two-thirds of models agree on the sign of the difference are stippled.

915





**Figure 7:** Time-pressure cross-sections of 5° N - 5° S mean zonal stratospheric winds (m s⁻¹) from the first ensemble member of each model's ssp245 simulation **(a – f)** and using ERA5 data **(g)**. The ERA5 plot was generated using Copernicus Climate Change Service information (Hersbach *et al*., 2019). Positive values indicate westerly winds and negative values indicate easterlies; the black contour is at 0 m s⁻¹.





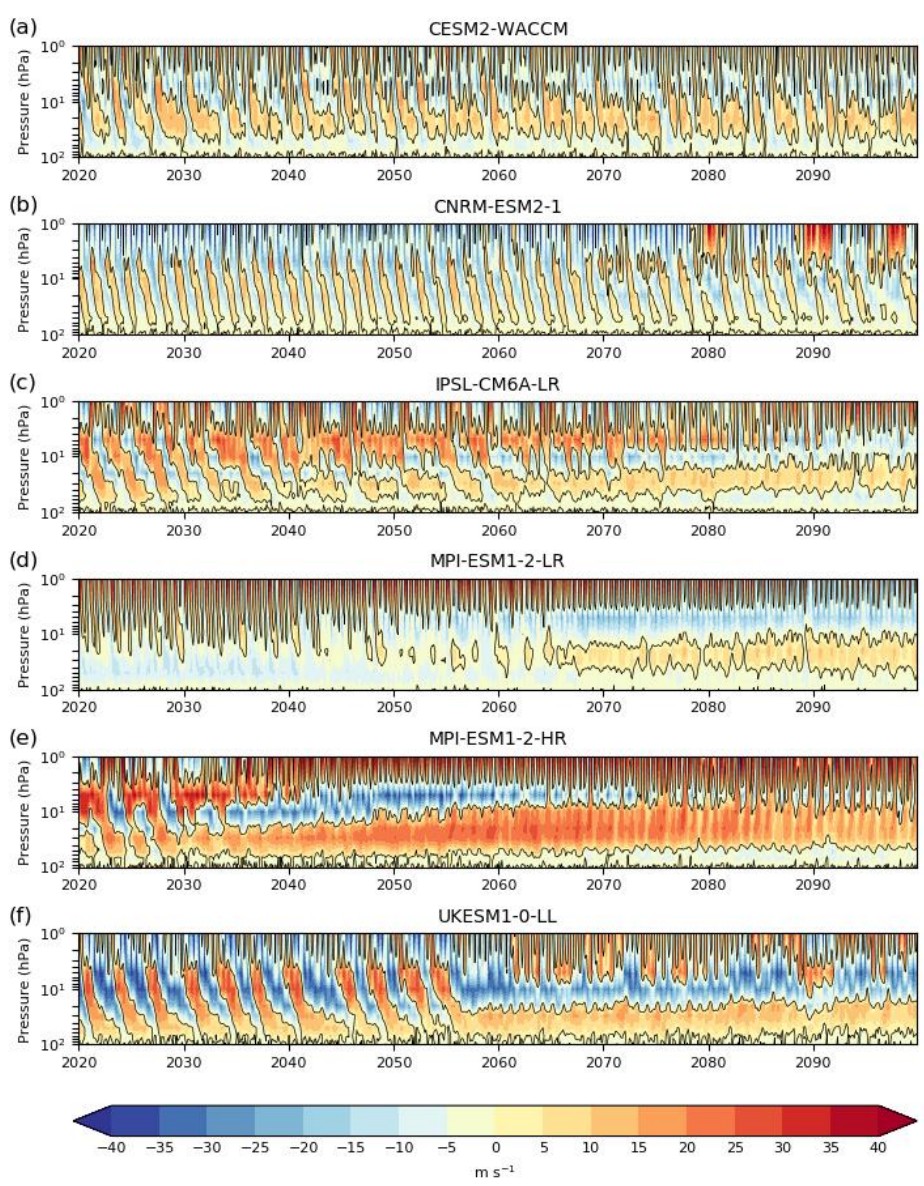

**Figure 8:** Same as Fig. 7 (a - f) but from the first ensemble member of each model's G6sulfur simulation.



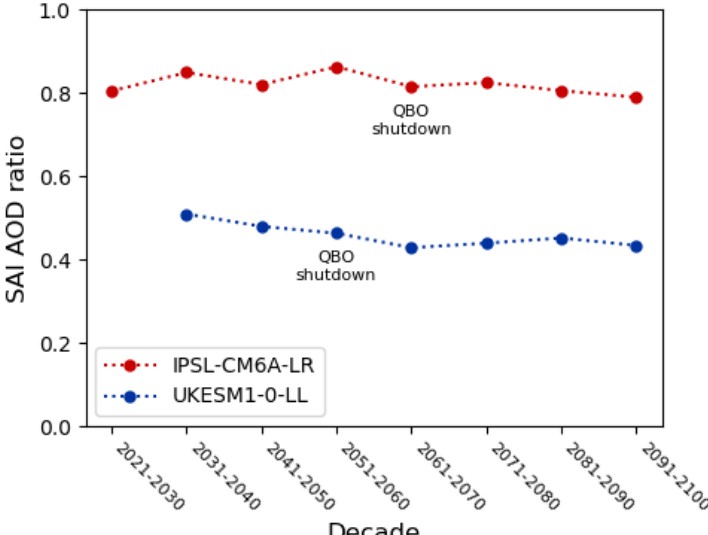

**Figure 9:** Evolution of the ratio of the mean SAI AOD in the subtropics to that in the SO$_2$ injection zone in ensemble member r1 of the two models whose G6sulfur experiments have both interactive SO$_2$ injection and a shutdown of the QBO. The injection zone is the area between 10° N & 10° S as specified in Kravitz *et al.* (2015) and the subtropics are defined here as the region between 30° N & 30° S but excluding the injection zone. SAI was not required in UKESM1-0-LL until 2031 so the decade 2021-2030 is omitted for this model.

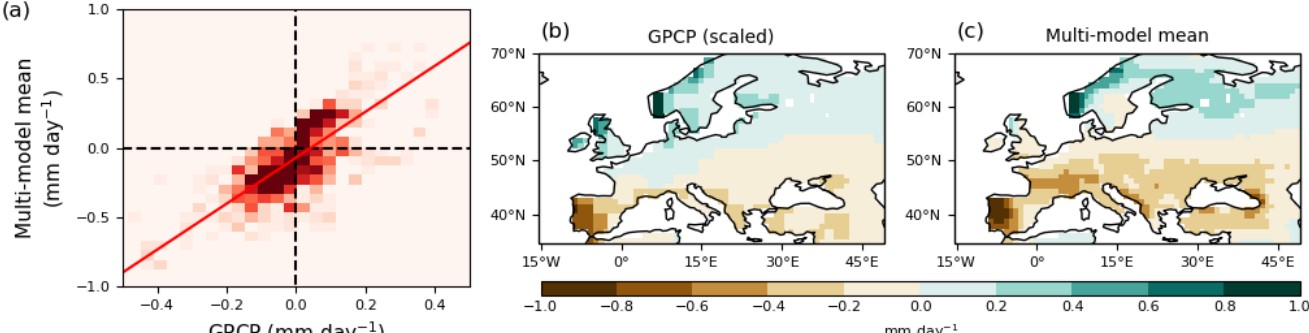

**Figure 10:** 2-D histogram and straight-line fit comparing the change in DJF land precipitation rate over Europe (mm day⁻¹) from GPCP observations with the multi-model mean; the GPCP data are the differences between positive and negative NAO winters over 1979-2015 and the model data are the mean differences between G6sulfur and ssp245 for 2081-2100 **(a)**. The difference in DJF land precipitation rate over Europe (mm day⁻¹) from GPCP as defined in panel (a) scaled by the gradient (1.65) of the straight-line fit in that panel **(b)**. The multi-model mean difference in DJF land precipitation rate over Europe (mm day⁻¹) between G6sulfur and ssp245 for 2081-2100 **(c)**.