# Peer review of "The impact of stratospheric aerosol intervention on the North Atlantic and Quasi-Biennial Oscillations in the Geoengineering Model Intercomparison Project (GeoMIP) G6sulfur experiment"

_Atmospheric Chemistry and Physics, 2021_

## Referee Comment (RC2)

**Review of *"The impact of stratospheric aerosol intervention on the North Atlantic and Quasi-Biennial Oscillations in the Geoengineering Model Intercomparison Project (GeoMIP) G6sulfur experiment"* by Jones et al.**

This is an excellent and very well written paper. The model intercomparison study, which investigates the impact of stratospheric aerosol injection (SAI) on the North Atlantic Oscillation (NAO) as well as on the Quasi-Biennial Oscillation (QBO), emphasizes regional impacts of SAI and compares scenarios with SAI and climate change (G6sulfur) to scenarios only with climate change and no SAI (SSP245 and SSP585). This is exactly the kind of comparison which needs to be made and analyzed to evaluate a potential future deployment of SAI. To me, the highlight of the paper is figure 6 which underlines the potential danger of regionally exacerbated precipitation (e.g., in Iberia) due to a forced positive winter NAO.

I recommend this study for publication in ACP. I agree with the comments made by reviewer 1 and I think that they should be processed before publication. The same goes for my additional minor comments and questions below.

**General Comment:** Despite the wide use of the term «sulphate aerosol» in the geoengineering community, it is mostly used wrongly, as it is the case in this paper. A sulphate normally refers to a salt containing $SO_4^{2-}$, but not liquids. Thus, the term "sulfuric acid aerosols" is more appropriate since the aerosols consist of a $H_2SO_4$-$H_2O$ solution.

**Line 66:** I recommend writing "correlated" instead of "associated", or do Zanardo et al., 2019 really show association?

**Line 117-119:** How can consistency be guaranteed here? Differences in radiative forcing do result in very different temperature responses across GCMs.

**Line 120-129:** The very different emission locations, especially differences in emission altitude, can cause significant differences in the distribution of the aerosol burden in the stratosphere and therefore very different SW cooling patterns as well as very different stratospheric heating patterns (e.g., Niemeier et al., 2009). How does this impact your results? This could be relevant, especially with respect to the QBO responses of SAI (see also Franke et al., 2021). This point should also be discussed in section 3.3.

**Line 165:** How does the NAO change in the SSP585 scenarios in the set of models considered in this study? The set of models in this study is different from the one in Tsanis and Tapoglou, 2019 (different set of models as well as model versions). How do they differ with respect to the NAO response?

**Line 163-168:** Doesn't G6solar (Kravitz et al., 2015) include the "confounding factor of different temperature" as well? The temperature difference is only due to solar constant reduction instead of SAI…

**Line 168:** You should make clear that the data referred to here is from your simulation and not Kravitz et al., 2015 as it was referred in the sentence before.

**Line 184-198:** This is very impressive. It would be great to see figure 6 also for temperature, not only for precipitation.

**Table 1:** The identifiers of the individual simulations are very long and confusing to distinguish. Could this be simplified?

**References:**

Franke, H., Niemeier, U., and Visioni, D.: Differences in the quasi-biennial oscillation response to stratospheric aerosol modification depending on injection strategy and species, Atmos. Chem., Phys., 21, 8615-8635, https://doi.org/10.5194/acp21-8615-2021, 2021.

Niemeier, U., Schmidt, H. and Timmreck, C. (2011), The dependency of geoengineered sulfate aerosol on the emission strategy. Atmosph. Sci. Lett., 12: 189-194. https://doi.org/10.1002/asl.304

Tsanis, I., and Tapoglou, E.: Winter North Atlantic Oscillation impact on European precipitation and drought under climate change, Theor. & App. Clim., 135, 323-330, https://doi.org/10.1007/s00704-018-2379-7, 2019.

Zanardo, S., Nicotina, L., Hilberts, A. G. J., and Jewson, S. P.: Modulation of economic losses from European floods by the North Atlantic Oscillation, Geophys. Res. Lett., 46, 2563-2572, https://doi.org/10.1029/2019GL081956, 2019.

---

## Author Comment (AC1)

**Response to Anonymous Referee #1**

We thank the Referee for their comments which have helped improve the paper. We quote the Referee's comments below (in **bold**) and then provide our responses to them; revised text is shown in red.

**Major comments**

1. **The definition of the NAO is unusual. Normally the NAO is defined as the difference of normalized (to unit variance) pressure in the two regions giving a dimension-less index. Here it is defined without the normalization. I see that the same definition is used in Stephenson et al. 2006 and Baker et al. 2018, but I did not find any motivation in these papers. The authors should describe the background for choosing a non-standard definition. They should also describe if the conclusions differ when using the more standard method.**

We follow Stephenson *et al*. (2006) and Baker *et al*. (2018)'s definition of the NAO because of its simplicity: being just the pressure difference between two regions makes it extremely easy to understand. Using a more complex definition of the NAO does not, however, affect our conclusions. Below we first show Fig. 2 from the manuscript using the simple definition of Stephenson *et al*. and Baker *et al*., followed by another version of the figure using a more complicated approach to define a dimensionless NAO index (e.g., Tsanis and Tapoglou, 2019; Hurrell *et al*., 2020):

[Figure]

The more complex definition first requires calculating the long-term mean and standard deviation for each region's DJF pressure timeseries, then the normalization of each series by first subtracting the long-term mean from each and then dividing by its standard deviation. The difference between the resulting series is a dimensionless NAO

index (which can have values outside the [-1, 1] range). Both versions of this figure show that the NAO is essentially unchanged in ssp245 over the period while exhibiting a clear positive trend in G6sulfur.

We have modified the text (lines 153-154) to state why we use the simple NAO definition and note that this does not affect our conclusions:

We use this definition of the NAO for its simplicity, but our conclusions are not affected by the use of a more complex NAO definition (e.g., Tsanis and Tapoglou, 2019; Hurrell *et al.* 2020).

References

Hurrell, J., and National Center for Atmospheric Research Staff (Eds): *The Climate Data Guide: Hurrell North Atlantic Oscillation (NAO) Index (station-based),* last modified 24 Apr 2020, available at: https://climatedataguide.ucar.edu/climate-data/hurrell-north-atlantic-oscillation-nao-index-station-based (accessed: 21 December 2021), 2020.

Tsanis, I., and Tapoglou, E.: Winter North Atlantic Oscillation impact on European precipitation and drought under climate change, *Theor. & App. Clim.*, **135**, 323-330, https://doi.org/10.1007/s00704-018-2379-7, 2019.

2. **In Figs. 5 and 6 the multi-model means are shown. The intermodel consistency -- significance of a non-zero signal? -- is calculated as where at least 4 out of the 6 models agree on the sign. If I understand this correctly, then even for random signs this agreement will happen with a probability of more than 50 %. The authors should estimate the significance with a more strict method.**

Figures 5 and 6 have been recreated using a two-tailed Student's *t*-test to define areas where differences are significant at the 5% level; the figure captions have been changed accordingly.

**Minor comments**

1. **l163: In order to assess .. I find this sentence rather convoluted. Why not compare directly to the ssp585 experiments?**

Experiment ssp585 cannot be used for comparison as it is much warmer than G6sulfur and this has been found to affect the NAO as noted in the manuscript. We have re-written the text (lines 164-170) to make this clearer:

In order to assess whether the NAO changes seen in G6sulfur are due to SAI the results from G6sulfur need to be compared against those from a similar experiment which does not include SAI and which also follows the same temperature evolution to G6sulfur, thus ruling out a straightforward comparison against ssp585. The latter condition is required because studies of scenarios with warming levels similar to ssp585 have been found to affect the NAO (e.g., Tsanis and Tapoglou, 2019). Both conditions are satisfied by GeoMIP experiment G6solar (Kravitz *et al.*, 2015) which is parallel to G6sulfur but achieves the cooling from ssp585 to ssp245 levels by the highly idealised method of reducing the specified solar output.

2. **l183: Christiansen 2018 (10.1175/JCLI-D-17-0197.1) explains why the model mean is better than individual models and could be cited here. As mentioned above, I don't think the inter-model consistency shows much.**

We thank the Referee for the reference and have included the citation at line 186.

3. **l225: In general the QBO will probably more or less vanish in an average of many experiments as its phase is almost random.**

We agree and is why we show the results from a single ensemble member and not an ensemble average.

4. **Given the length of the paper it contains many figures. Perhaps a few of them could be discarded (maybe Fig. 10).**

We would prefer to keep the current figures as we believe they are useful.

---

## Author Comment (AC2)

**Response to Anonymous Referee #2**

We thank the Referee for their kind words and comments. We quote the Referee's comments below (in **bold**) and then provide our responses to them; revised text is shown in red.

**General Comment**

**Despite the wide use of the term «sulphate aerosol» in the geoengineering community, it is mostly used wrongly, as it is the case in this paper. A sulphate normally refers to a salt containing $SO_4^{2-}$, but not liquids. Thus, the term "sulfuric acid aerosols" is more appropriate since the aerosols consist of a $H_2SO_4$-$H_2O$ solution.**

We have made the suggested change throughout the manuscript.

**Specific Comments**

1. **Line 66: I recommend writing "correlated" instead of "associated" or do Zanardo et al., 2019 really show association?**

   We have made the suggested change (line 66).

2. **Line 117-119: How can consistency be guaranteed here? Differences in radiative forcing do result in very different temperature responses across GCMs.**

   We agree that differences in radiative forcing can indeed result in very different temperature responses across GCMs. This issue is avoided in these GeoMIP simulations by specifying the criterion for consistency in terms of temperature response rather than radiative forcing: all participating models are consistent in that their G6sulfur simulations have cooled sufficiently to match the temperature evolution of their respective ssp245 simulations, as described in lines 116-119.

3. **Line 120-129: The very different emission locations, especially differences in emission altitude, can cause significant differences in the distribution of the aerosol burden in the stratosphere and therefore very different SW cooling patterns (e.g., Niemeier et al., 2009). How does this impact your results? This could be relevant, especially with respect to the QBO responses of SAI (see also Franke et al., 2021). This point should also be discussed in section 3.3.**

   We agree that the altitude at which aerosol is introduced can affect its radiative impact as shown by Niemeier *et al*. Nevertheless, all model perturbations are still in the equatorial lower stratosphere even if the exact locations differ, and all models achieve the GeoMIP goal of cooling their ssp585 configurations to ssp245 temperatures via SAI as required. Assessing the impact of different SAI emissions locations is beyond the scope of this study and would require a new intercomparison effort. As we note in Section 3.3 there is no correlation between the amount of stratospheric warming induced in the models with whether or not the QBO shuts down. We have now made this point more explicitly (lines 242-244) and now also include the Niemeier *et al*. reference (from 2011 as indicated in the Referee's reference list, not 2009 as used in the comment):

   Although the altitude of $SO_2$ injection can affect the radiative impact of SAI (e.g., Niemeier *et al*., 2011), there is no correlation between the amount of stratospheric warming induced by SAI and the shutting down of the QBO…

4. **Line 165: How does the NAO change in the SSP585 scenarios in the set of models considered in this study? The set of models in this study is different from the one in Tsanis and Tapoglou, 2019 (different set of models as well as model versions). How do they differ with respect to the NAO response?**

The multi-model mean gradient in ssp585 over 2021-2100 is +0.30 hPa decade$^{-1}$. This cannot be compared quantitively with the results of Tsanis and Tapoglou (2019), nor would we expect quantitative agreement given the different models/versions and scenarios as noted by the Referee, but in a qualitative sense both indicate a positive NAO trend in high-forcing/warming scenarios (SSP5-8.5 in our study, RCP8.5 in theirs) – this is the only point being made here. We have modified the text to include the result from ssp585 (line 172-173):

For reference, the multi-model mean gradient in ssp585 is +0.30 hPa decade$^{-1}$ (range -0.22 to +0.86 hPa decade$^{-1}$).

5. **Line 163-168: Doesn't G6solar (Kravitz et al., 2015) include the "confounding factor of different temperature" as well? The temperature difference is only due to solar constant reduction instead of SAI...**

By design, experiments ssp245, G6sulfur and G6solar all follow the same time-profile of global-mean temperature whereas ssp585 does not – ssp585 is warmer. Although they did not investigate exactly the same scenarios we use here, Tsanis and Tapoglou (2019) have shown that such warming in itself causes a change in the NAO, a finding our results confirm (see response to Comment 4 above). By restricting the comparison to ssp245, G6sulfur and G6solar this warming-induced source of NAO change is avoided. The text has been revised as follows (lines 164-170) to make this clearer:

In order to assess whether the NAO changes seen in G6sulfur are due to SAI the results from G6sulfur need to be compared against those from a similar experiment which does not include SAI and which also follows the same temperature evolution to G6sulfur, thus ruling out a straightforward comparison against ssp585. The latter condition is required because studies of scenarios with warming levels similar to ssp585 have been found to affect the NAO (e.g., Tsanis and Tapoglou, 2019). Both conditions are satisfied by GeoMIP experiment G6solar (Kravitz *et al.*, 2015) which is parallel to G6sulfur but achieves the cooling from ssp585 to ssp245 levels by the highly idealised method of reducing the specified solar output.

6. **Line 168: You should make clear that the data referred to here is from your simulation and not Kravitz et al., 2015 as it was referred in the sentence before.**

We have amended the text to make this clearer (line 170):

The multi-model mean NAO gradient of the G6solar simulations from the models used here is…

7. **Line 184-198: This is very impressive. It would be great to see figure 6 also for temperature, not only for precipitation.**

We have expanded Fig. 6 to include the temperature response and have modified the associated text as follows (lines 190-197):

We now examine temperature and precipitation over Europe to compare changes from present-day under scenario SSP5-8.5 plus SAI (i.e. G6sulfur) with the situations under scenarios SSP2-4.5 and SSP5-8.5. Figure 6(a) shows the multi-model-mean difference in DJF mean surface air temperature between end-of-century in ssp245 (mean over 2081-2100) and present-day (mean over 2011-2030) which shows a typical pattern of increased warming at higher latitudes and over land. As expected, this warming is

greater in ssp585 (Fig. 6b) and the latter is then somewhat reduced in G6sulfur (Fig. 6c) while also showing the impact of the change in the NAO. Figures 6(d-f) show the corresponding multi-model-mean differences in DJF mean land precipitation rate. In ssp245 (Fig 6d) there is a general increase in precipitation over most of central and northern Europe and a slight reduction over southern Europe....

The panel identifiers have been changed in the rest of the text and the figure caption revised.

8. **Table 1: The identifiers of the individual simulations are very long and confusing to distinguish. Could this be simplified?**

We agree that the identifiers are somewhat unwieldy but they are the official identifiers used by CMIP6 (we have not invented them) and are what is required to identify specific simulations in the ESGF archive.